# Evaluating theory of change to improve the functioning of the network for improving quality of care for maternal, newborn and child health

Albert Dube[1,2]*, Kondwani Mwandira[1,2], Kohenour Akter[3], Fatama khatun[3], Seblewengel Lemma[4], Gloria Seruwagi[5], QCN Evaluation Group[¶], Yusra Ribhi Shawar[6,7], Nehla Djellouli[8], Charles Mwakwenda[3], Mike English[9], Tim Colbourn[8]*

1 Parent and Child Health Initiative Trust (PACHI), Lilongwe, Malawi, 2 Malawi Epidemiology and Intervention Research Unit (MEIRU), Lilongwe, Malawi, 3 Perinatal Care Project, Diabetic Association of Bangladesh, Dhaka, Bangladesh, 4 London School of Hygiene & Tropical Medicine, London, United Kingdom, 5 School of Public Health, Makerere University, Kampala, Uganda, 6 Bloomberg School of Public Health, John Hopkins University, Baltimore, Maryland, United States of America, 7 Paul H. Nitze School of Advanced International Studies, John Hopkins University, Washington D.C., United States of America, 8 Institute for Global Health, University College London, London, United Kingdom, 9 Centre for Tropical Medicine and Global Health, University of Oxford, Oxford, United Kingdom

¶ Membership of the QCN Evaluation Group is listed in the Acknowledgments.
* albert.dube@meiru.mw (AD); t.colbourn@ucl.ac.uk (TC)

**Data Availability Statement:** The original qualitative data are not made available since some responses contain potentially identifiable

## Abstract

In 2017, WHO and global partners launched 'The Network for Improving Quality of Care for Maternal, Newborn and Child Health' (QCN) seeking to reduce in-facility maternal and newborn deaths and stillbirth by 50% in health facilities by 2022. We explored how the QCN theory of change guided what actually happened over 2018–2022 in order to understand what worked well, what did not, and to ultimately describe the consequences of QCN activities. We applied theory of change analysis criteria to investigate how well-defined, plausible, coherent and measurable the results were, how well-defined, coherent, justifiable, realistic, sustainable and measurable the assumptions were, and how independent and sufficient the causal links were. We found that the QCN theory of change was not used in the same way across implementing countries. While the theory stipulated Leadership, Action, Learning and Accountability as the principle to guide network activity implementation other principles and varying quality improvement methods have also been used; key conditions were missing at service integration and process levels in the global theory of change for the network. Conditions such as lack of physical resources were frequently reported to be preventing adequate care, or harm patient satisfaction. Key partners and implementers were not introduced to the network theory of change early enough for them to raise critical questions about their roles and the need for, and nature of, quality of care interventions. Whilst the theory of change was created at the outset of QCN it is not clear how much it guided actual activities or any monitoring and evaluation as things progressed. Enabling countries to develop their theory of change, perhaps guided by the global framework, could improve

information at multiple levels including for individuals, health facilities, and organizations; the Institutional Review Board conditions stipulated that participants will not be at risk of identification through their responses; participant consent forms did not include obtaining explicit consent regarding the possible use of anonymized data in the public domain via a data repository. Non-author contact information for enquiries about data access: please email dataprotection@ucl.ac.uk.

**Funding:** This work was funded by the Medical Research Council (MRC) Health Systems Research Initiative 5th call via grant MR/S013466/1 to TC at UCL Institute for Global Health, United Kingdom, YS and JS at Johns Hopkins University, United States of America, KA and AK at Diabetic Association of Bangladesh Perinatal Care Project, Bangladesh, CM at Parent and Child Health Initiative, Malawi, GS at Makerere University School of Public Health, Uganda, and ME at University of Oxford, United Kingdom; and by the Bill & Melinda Gates foundation via grant INV-007644 to TM at LSHTM, United Kingdom. The funders had no role in study design, data collection and analysis, decision to publish, or preparation of the manuscript.

**Competing interests:** The authors have declared that no competing interests exist.

stakeholder engagement, allow local evaluation of assumptions and addressing of challenges,, and better target QCN work toward achieving its goals.

## Introduction

Far too many women and babies continue to die from complications in pregnancy and childbirth due to poor quality of and access to peripartum care, especially in low and middle income countries. In 2020, over 2.4 million newborn babies and an estimated 287 000 women died due to lack of quality of care at or immediately after child birth and in the first days of newborn life [1–6]. In 2017, WHO and global partners launched 'The Network for Improving Quality of Care for Maternal, Newborn and Child Health' (QCN) [7] seeking to reduce in-facility maternal and newborn deaths and stillbirth by 50% in health facilities by 2022, initially in nine countries: Bangladesh, Cote d'Ivoire, Ethiopia, Ghana, India, Malawi, Nigeria, Tanzania and Uganda. Their aim was to build a cross-country platform for joint-learning around quality improvement implementation approaches based on a shared theory of change and shared health outcome goals. The QCN operated at global, national and local levels, facilitating implementation of a range of quality improvement efforts for different aspects of maternal, newborn and child healthcare, starting in selected 'learning facilities' in 'learning districts' and scaling up to other health facilities and districts [7].

Although there is an emerging body of work on networks as a potential change strategy in both low- and middle-income countries and high-income countries [8–11], interventions to improve quality of care for maternal, newborn and child health are complex and require comprehensive and detailed theories of change that contribute nuanced understanding of how to address evidence-to-practice gaps and ultimately contribute to better health outcomes [12]. Investigating assumptions and causal linkages set out in theories of change through evaluation can also contribute to better design and implementation of programmes, and refined theories of change.

Theories of change can play a critical role during the formulation and design of programmes and can as well enable monitoring and evaluation of programmes that is responsive to stakeholder needs [13]. Theories of change have been used as frameworks for designing simple and complex interventions. Project and programme stakeholders have utilised theories of change as strategic plans [14–16]. Theories of change have also been at the centre of public health intervention evaluation. For example, programme managers and evaluation teams have used theories of change in the development of process as well as outcome indicators for monitoring of implementation [17], formulation of evaluation questions, testing assumptions [16, 18, 19] and exploration of the influence of programme context on performance [16]. Theories of change can also be used in the identification of implementation or programme failures, side effects, effectiveness and causal explanations [20].

In 2016–7, WHO, governments and key international development partners developed a cross-country theory of change for the quality of care network for maternal and newborns. The theory of change was developed by global level stakeholders of the QCN (led by the WHO and UNICEF core group) from September 2016 to February 2017 and shared with 340 national stakeholders and partners at the launch of the network in Lilongwe, Malawi in February 2017 [21]. The theory highlights intervention themes and processes that are necessary for the attainment of clinical and behavioural outcomes that lead to improved quality and outcomes of care for women and newborns. The theory of change further presents socioeconomic status, gender

**Cross-cutting Dimensions**

- Socioeconomic status: Does this reduce access to good quality care among the poorest? Reduce inequality?

- Gender: Do women staff and users feel involved and empowered in each component of the intervention? Does it promote gender equity in access to care and other resources?

- Resilience and sustainability: Does the intervention improve resilience, organizational culture and resistance to shocks over time?

**Improved outcomes for women, newborn infants and children**
**Survival, less morbidity, user satisfaction and dignity**

Women's and children health outcomes

**Improved quality of care**

**Improved access to care for marginalised groups**

**Improved dignity and satisfaction**

Quality of care outcomes

**Changes in services and referral practices**
- Improved teamwork
- Improved skills/signal functions
- Improved quality improvement processes

**Increased individual and community empowerment**
- Improved social support/capital
- Improved wellbeing
- Empowerment and improved equity in decision making

**Improved user satisfaction**
- Improved environment for childbirth
- Improved maternal and infant practices (pre-and post-partum)
- Changes in care-seeking, coping behavior and in user satisfaction

Clinical and community behavioral outcomes

**Structural capital**
- Improved WASHE at facilities
- Improved hygiene practices
- Improved energy supply at facilities

**Financial capital**
- Improved ability to cover cost of care
- Improved access to health care

**Social Capital**
- Improved sharing of knowledge and materials between providers
- Changed norms around management and quality improvement practices

**Human capital**
- Knowledge of best quality of care practices by service providers
- Improved facility management skills for service and dignity
- Improved gender equity, women empowerment and dietary practices

Intervention processes

**Integrated quality Improvement by participatory Teams**

| **Environment** | **Leadership** | **Action** | **Learning systems** | **Accountability** |
|---|---|---|---|---|
| Managing care facilities for hygiene and dignity | Country-led | Standards and resources | Data systems | National framework |
| Organizational culture | Structures | Phased implementation | Audit/team meetings | Institutionalization |
| Political commitment | Plans | Institutionalization | FDSA cycles and PLA | Evaluation, internal and external |
| | Mobilization | | Global learning framework | |

Intervention themes
i. National teams
ii. District teams
iii. Facility teams

**Fig 1. Quality of care network theory of change as developed by WHO.**

and resilience and sustainability as cross-cutting influences (Fig 1). The theory of change was used by QCN stakeholders primarily as a logical framework to guide areas of implementation, and to guide monitoring and evaluation of the network including via comparison of progress between countries [7, 21].

The aim of this paper is to explore how the quality of care network theory of change guided what actually happened over 2018–2022 in order to understand what worked well, what did not, and to ultimately help explain the consequences of QCN activities [22].

## Methods

We conducted an evaluation of the global quality of care network focusing on four implementing countries as case studies: Malawi, Bangladesh, Ethiopia and Uganda [23]. Our main research questions were around the emergence [24], legitimacy [25], and effectiveness of the network [26]. We also conducted a social network analysis of the quality of care network [27], evaluated the influence of contextual factors at individual, organisational and institutional levels on the functioning of the network [28], looked at the learning and innovation aspects of the network [29], and its sustainability [23] (also see S1 Text for an overview of QCN evaluation work). The main findings of our evaluation were that global and national leadership elements

**Table 1. Qualitative interviews and health facility observations completed, by time, and country.**

| Case-study Country | Data collection dates | National interviewee (n) | Sub-national Interviewee (n) | Facility Observation (n) |
|---|---|---|---|---|
| Bangladesh | 1 (Oct-2019 –Mar-2020) | 13 | 7 | 3 |
| | 2 (Oct -2020 –Jan-2021) | 14 | 11 | 4 |
| | 3 (May-2021 –Sep-2021) | 10 | 12 | 4 |
| | 4 (Jan-2022 –Mar-2022) | 8 | 0 | 0 |
| Ethiopia | 1 (Dec-2020– Mar-2021) | 8 | 11 | 4 |
| | 2 (Sep-2021 –Dec-2021) | 10 | 11 | 3 |
| Malawi | 1 (Oct-2019 –Mar-2020) | 7 | 12 | 4 |
| | 2 (Nov-2020 –Jan-2021) | 10 | 7 | 4 |
| | 3 (Aug-2021 –Nov-2021) | 9 | 7 | 4 |
| | 4 (Mar-2022 –May-2022) | 4 | 3 | 0 |
| Uganda | 1 (Nov-2020 –Mar-2021) | 7 | 13 | 4 |
| | 2 (June-2021 –Sep-2021) | 12 | 8 | 4 |
| | 3 (Feb-2022 –Mar-2022) | 10 | 5 | 4 |

of QCN have been most effective to date; action, learning and accountability have been more challenging, partner or donor dependent, remaining to be scaled-up, and pandemic-disrupted. We draw on all of these studies as well as bespoke analyses for this paper specifically investigating how and why aspects of the network worked, or did not. Iterative rounds of interviews were conducted with key stakeholders such as QCN coordinators, UNICEF, WHO, UNFPA, and GIZ national representatives, Ministry of Health monitoring and evaluation officers, directors of health and social services at each learning facility, and quality improvement officers in each country at both national and local level. We reviewed accessible published and unpublished documents and communications relating to the quality of care network at global level and at national and local levels in each of the four countries. These included strategic plans and management documents, operational plans, directives, formal minutes, and reports. We conducted non-participant observations of multi-country, national-level and district level meetings in case-study countries. Activities at district level were also observed; including visits to two better and two least performing 'learning facilities' in each of the four countries in several iterative rounds. Table 1 provides an overview of the data we collected.

We adapted a psychometrically validated tool developed for evaluating clinical networks to evaluate the network at national and local levels in Bangladesh, Ethiopia, Malawi and Uganda (Appendix 5 in S2 Text). We conducted several rounds of the survey in each country between October 2019 and March 2022 and in each round a wide variety of network member cadres (clinicians, managers, and advisors) were surveyed. A detailed report on the methodology of the whole quality of care network evaluation is presented in S2 Text, and we provide reflections on our experiences of evaluating the quality of care network in another paper in our QCN Evaluation collection by Gloria Seruwagi and colleagues [30].

### Theory of change analysis

We applied theory of change analysis criteria developed by John Mayne [31]. We did an ex-post analysis in order to appraise the whole theory of change. We looked at results, assumptions and causal pathways. We investigated how well-defined, plausible, coherent and measurable the results were, how well-defined, coherent, justifiable, realistic, sustainable and measurable the assumptions were, and how independent and sufficient the causal links were. Findings from this analysis were then used to explore how the quality of care network theory of change guided what actually happened over 2018–2022 in order to understand what worked

**Table 2. Theory of change analysis data extraction tool.**

| Resources / Stakeholders | Country Name | | | | | | | |
|---|---|---|---|---|---|---|---|---|
| | Activity | Changes at community level | Changes at facility level | Changes at national level | Outcome at National level | Pre-condition | Assumptions | Challenges |
| | | | | | | | | |

well, and what did not, in each country, and globally [31]. A data extraction tool was used to pull together data from quality of care network survey reports, qualitative findings, facility observation reports, meeting observation reports, strategic documents and our other seven papers evaluating the quality of care network [23–29]. This data extraction tool, Table 2, was refined from Daruwalla 2019 [32] to mine resources and stakeholders, quality of care network activities, changes at community level, changes at facility level, and outcomes at national level, documentation of pre-conditions, assumptions and challenges.

## Ethics

Ethical approval was received from University College London Research Ethics Committee (ref: 3433/003); BADAS Ethical Review Committee (ref: BADAS-ERC/EC/19/00274), Ethiopian Public Health Institute Institutional Review Board (ref: EPHI-IRB-240-2020), National Health Sciences Research Committee in Malawi (ref: 19/03/2264) and Makerere University Institutional Review Board (ref: Protocol 869). The conduct of the evaluation was based on clear ethical standards which assured confidentiality, privacy, anonymity and informed consent. All respondents provided verbal or written informed consent. All respondents were informed of: (i) the purpose of the evaluation; (ii) their right to refuse to participate; and (iii) that their possible decision not to participate would not be held against them or affect their status in the network.

## Results

In this section, we present the results of our analysis of the global theory of change for a quality of care network for improving maternal, newborn and child health in low and middle income countries focusing on four countries: Malawi, Uganda, Bangladesh and Ethiopia. We first present the overall understanding of the network. This is followed by theory of change intervention themes, processes, and clinical and behaviour outcomes. This section ends with a presentation of aspects that we found to be missing in the theory of change even though we found them to be key in the achievement of QCN results in specific countries.

### Awareness, understanding and owning the network

We found that governments and national level stakeholders in Bangladesh, Ethiopia, Malawi and Uganda were aware of the network history, mission and its organisation structure while awareness was limited amongst implementers at sub-national and facility level. Several respondents reported:

> "The other problem is a national problem in Ethiopia in general, particularly in our region, it (QCN) was considered as a campaign work. I thought it is just one or two months' work. I understood the details later on. It's after reviewing different documents about the network, that I understood it is a global initiative" (Ethiopia-05-facility staff)

*"Of course, they [Quality of care network activities] were there but doing it unknowingly, you would find that you are doing work very nicely but you are not aware, you are not appreciating yourself like you have done this one."* (Uganda-02-facility staff)

*"We still need to do capacity building to the leaders. I am saying this because the leaders or management team of the health bureau don't have sufficient awareness about quality of care. When I join the health bureau as a quality officer, I have clearly observed this and it has been a serious challenge."* (Ethiopia-07-national partner)

It was also found however that awareness at learning facilities and sub-nationally increased over time.

We found that, much as the network was a government initiative within each country, funding and technical support for QCN activities was provided to national level QCN implementers by different national level NGO implementing partners. This resulted in some sub-national level actors perceiving QCN activities of the network as partner's work rather than looking at the QCN as a government-driven initiative. For instance:

*. . .these significant national-level partners, which could be best classified as "change agents" within the Bangladesh system, are Save the Children-IHI with their MaMoni project, and UNICEF, with its EMEN project. . .A further "change agent" within the system is QIS/HEU, which holds operational planning control over many of the country's quality projects* (Bangladesh-02-National partner).

Similarly, another respondent noted:

*. . . But you see USAID is everywhere, in almost all these regions of the country. We have Mbale, Lira, Gulu, South West Mbarara has like 16 districts, and in fact UNFPA is doing some good work in the Karamoja region with around 9 districts. Then UNICEF has supported 3 JPHIEGO about 13 districts in the west Nile region and we have USAID in Lira has like 5 districts, Gulu has like 7 districts. But this is something we are telling the Ministry of Health, can they now start to have a map for these activities so that we can tell where these activities are happening?* (Uganda-03-National partner).

This perception was however transformed progressively, though not entirely, through meetings, workshops, and training organised by implementing partners, and when health workers at facilities started to comprehend the benefits. Some participants perceived the network vision, objectives and standards as being more of the higher up agenda which rarely takes into consideration all local resource environments and realities as noted here:

*. . .. The standards are great but the variation is in the resource envelope. . . ..Uganda's playground is bushy and there are no resources to clear and remove all. So, sometimes that's the challenge with global goals. They are high up there and the ground is not flat but we always travel without coming up with excuses to aim at achieving the same.* (Uganda-03- National partner)

### QCN theory of change: Intervention themes

**Integrated quality improvement by participatory teams.** The Network for Improving Quality of Care for Maternal, Newborn and Child Health (QCN) was designed to take an

integrated approach to quality improvements by participatory teams by working through learning, action, leadership and accountability (LALA) in the context of the facility management, organisation culture and political commitment. We found variations in the approaches that counties took to implement QCN activities. For instance, Plan, Do, Check and Action (PDCA) and 5S (Sort, Straighten, Shine, Standardise and Sustain). These were used in Bangladesh and Malawi respectively.

*"Currently, most of the Quality improvement teams where these WITs* [Work Improvement Teams] *are attached to, have done the five S (sort, set in order, shine, standardize, sustain the cycle) activities". (*Malawi-01-National partner)

In addition to different implementation approaches adopted by implementing countries, actors working at different levels of the health system reported poor documentation practices; poor quality of data collected; lack of data collection tools; lack of knowledge on how to use them; and lack of a culture of data use for decision making, including parallel reporting systems for QNC related indicators.

*"The other thing is the parallel reporting system, which is our biggest challenge. This is not included in DHIS 2. Extracting these 15 common core indicators from the chart is a big challenge. . .most of all; the parallel reporting system is our biggest challenge" (*Ethiopia 04-National partner).

These differences in approaches are suggestive of the fact that leadership, learning, action and accountability assumptions are not similar across implementing countries. We found that these assumptions are not explicitly put in the theory of change. Such approaches were linked to different implementation methodologies of key stakeholders already implementing quality improvement activities in health facilities.

**Intervention processes: Structural, financial, social and human capital.** The theory of change for the network presents four pillars of process level changes that are necessary to effect clinical and community level outcomes. We found that countries were at different levels in terms of the status of these pillars. QCN initiated trainings were perceived to have improved the availability of skilled professionals in learning facilities however not all skilled staff provided care, facilities experienced high levels of staff attrition and a demotivated workforce due to high demand for services. One respondent noted

*"The second challenge is also the continuous dropping out of trained workers to be replaced by new workers. The new employees are not trained and take additional time to fill the skill gap."* (Bangladesh-05 sub-national partner)

One of the assumptions of the network was that knowledge and learning would be shared among network member health professionals, and periodic learning forums were organised for this purpose in order to strengthen the social capital in the provision of quality care. While, in Ethiopia and Malawi, the learning forums were considered very motivating for health workers, it was observed that some staff attending these forums competed over allowances, which affected attendance in some knowledge sharing meetings in Uganda.

We found that lack of physical resources–infrastructure, basic amenities, and equipment— were frequently mentioned that prevented adequate care or reduced patient satisfaction. Providers reported that this problem had tied their hands when exercising skills and knowledge they acquired through various quality of care trainings.

*". . .Most donors will say that is a responsibility of government and will focus largely on software but if the facility has a small maternity room where there is no privacy, no light, no water and government has no capacity to provide this, how can we even start to talk about experience of care?. . ."* (Uganda 02-Sub national partner)

## Clinical and behavioural change outcomes

Clinical and community behaviour change outcomes (yellow boxes in Fig 1) were expected to result from the successful intervention processes. We found limited reports on achievements of results at this level in the causal pathways. Table 3 presents key changes that were reported to be related to QCN activities in the participating countries. In countries where some results at outcomes and impact levels were reported, these relate to initiatives that predated the launch of quality of care network activities.

## Quality of care network implementation coverage, timelines and other omissions

Quality of care network activities were set to be implemented in learning sites and by 2019 all four countries had identified initial learning sites. With a total of 255 public hospitals and 5054 private hospitals, Bangladesh scaled up QCN activities to over 298 facilities by early 2022. During the same period with a total of 6937 health facilities, Uganda moved from 18 facilities to 88 facilities while no scale up was reported in Ethiopia which has a total of 353 public hospitals and 3706 public health centres and Malawi which has a total of 1724 facilities. Ethiopia's plan to scale up was disrupted by the Covid-19 pandemic. We also found differences in the acceleration of network activities and different levels of implementation across implementing countries.

Fig 2 presents key challenges experienced during the implementation of the quality of care network in the four study countries.

These challenges manifested differently in different countries and affected the network activities, learning and accountability. One of the aspects of the network that was affected by these challenges was the speed at which implementation could take place. In the period between 2017 and 2022, Bangladesh developed and popularised its operational plans and standard operating procedures while Ethiopia, Malawi and Uganda developed operational plans more slowly and by 2019 few sub-national and facility level actors were aware of these plans. In Uganda however, non-awareness of QCN plans was due to the fact that the ministry of health mainstreamed QCN

**Table 3. Changes reported to be related to quality of care network activities.**

| Malawi | Uganda |
|---|---|
| Adoption and addition of QCN* indicators<br>Increased use of postnatal check-list tool reduction in birth asphyxia in some facilities | Improved cooperation and cohesion amongst partners<br>Emphasis on evidence based interventions<br>Ability to measure MPDSRs* |
| **Bangladesh** | **Ethiopia** |
| Improved accountability and learning<br>Availability of EMEN* dashboard<br>Infrastructure improvements at facility level<br>Establishment of breastfeeding corners<br>Changed recording and reporting of indicators<br>Strengthened coordination of QI* activities | Improved the skill at facility level<br>Increased use of partograph<br>Improvement in infrastructure<br>Introduction of counselling service |

*EMEN: Every Mother Every Newbron; MPDSR: Maternal and Perinatal Death Surveillance and Response; QCN: The Network for Improving Quality of Care for Maternal, Newborn and Child Health; QI: Quality Improvement

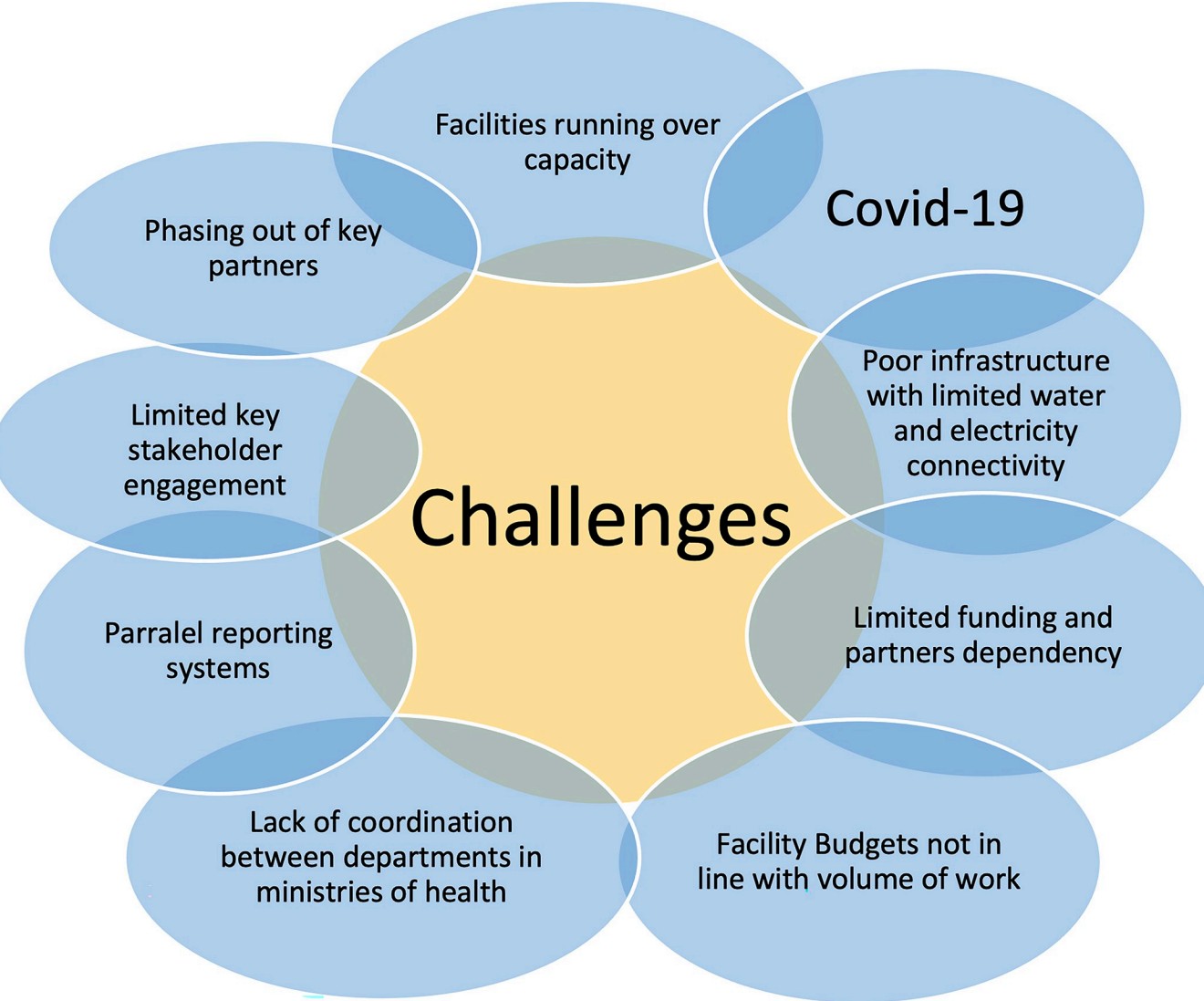

**Fig 2. Key quality of care network implementation challenges.**

activities into maternal newborn and child health and general quality improvement activities and as a result it was challenging for implementers to isolate QCN focused activities.

We found that infrastructure and its contribution to improvement in quality of care for women, newborn and children is missing in the theory of change. We also found that all countries reported challenges relating to structural, social and human capital. Reports on how QCN activities have contributed to improved ability to cover costs of care and improved access to health care were not common. It is not clear whether non-reporting on this pillar was due to a lack of clarity on who might cover costs or a lack of specificity as to whether the focus is on access to general health care or access to quality maternal, newborn and child health care.

## Discussion

The aim of this paper was to explore how the quality of care network theory of change–that primarily took the form of a high-level logic model–guided what actually happened over

2018–2022 in order to understand what worked well, and what did not. Studies have proposed a more comprehensive application of theory of change to the micro-and meso-levels of individuals, households and communities, where the results of impact investments matter most [33–35]. We found that the QCN theory of change was not specifically adapted for use at country level and was not used in the same way across implementing countries. While the theory stipulated Leadership, Action, Learning and Accountability as the the principles to guide network activity implementation, other principles and quality improvement methods, such as the 5 S and PDCA, have also been used. In addition to difference in implementation approaches we also found differences in reporting systems. While the majority of the countries integrated reporting on Network indicators through DHIS2, one country developed a parallel reporting system. Our findings are suggestive of the fact that while it might be useful to develop an overall theory of change for QCN it would then make more sense to use this as the starting point for country level theory of change development, that involves all relevant local stakeholders, to create a specific and more elaborate theory of change for the country context [13, 36].

We found strong consensus and partner engagement at global and national level. This was weaker among key stakeholders and network implementers at sub-national and facility levels to the extent that some were not aware of the network and its objectives, while others perceived the network as implementing partner activities rather than a ministry of health national initiative championed by national governments. The network and its theory of change were developed following a top-down approach leading to issues of ownership, adoption and accountability at local level. Key partners and sub-national and facility level implementers were not introduced to the network theory of change early enough for them to raise critical questions about their roles and the need for, and nature of, quality of care interventions [37].

It has been argued that many of the assumptions about how the world works are based on implicit theories of change, based on our worldview, developed through our education and upbringing [38]. This means that notwithstanding differences in worldviews, education levels and upbringing, differences in approaches to solving social challenges also prevail. These differences are external and outside the direct control of project management and implementers, but nevertheless should be addressed when developing programme theories of change [39]. We found that key conditions were missing at service integration and process levels in the global theory of change for the network. These missing elements suggest that national level differences were either remote to the developers of the theory of change or these country level differences could not all be used to inform its development. Our findings raise questions about how a high-level, global theory of change should inform, or can inform, national or sub-regional programmes. For example, conditions such as lack of physical resources that are locality dependent were frequently reported to be preventing adequate care, or harm patient satisfaction. Lack of physical resources tied the hands of trained service providers from exercising skills and knowledge they acquired through various quality of care training sessions. We also found that some pillars of the QCN theory of change, such as community empowerment, were rarely being worked on or at least not reported on, suggesting that such pillars were not seen as a priority in the local context.

The main strength of this study is that we were able to use data from multiple sources, over four years. This enabled us to triangulate and validate results from more than one source. The main limitation of our study is that we were not able to investigate all aspects of the theory of change in all countries. Although our analysis drew on a lot of primarily qualitative data (interviews, document review, a survey, observations), we were ultimately limited by the data we had and were not able to explore all assumptions and potential causal pathways. Our investigation was also limited by the original theory of change not explicating assumptions and mechanisms that might have informed our evalution, and by the QCN not having any in-built

evaluations at country level that themselves exposed and examined assumptions and mechanisms. This left some gaps in both our understanding, and the QCN team's understanding, of the programme. For example, in relation to whether, how, and how much, the QCN was able to improve access to care for marginalised groups, improve dignity and satisfaction of patients, and reduce mortality and morbidity in network learning facilities or more broadly.

In conclusion, our paper has provided an assessment of the global theory of change for a network on quality of care for maternal and newborn health. We found different levels of awareness of the network and its objectives, different approaches being used in different countries, absence of key conditions as reported by different implementing countries, and the presence of pillars in the global theory of change that are rarely reported on. Our review of implementation and coordination meeting reports, analysis of learning facilities observation reports, analysis of transcripts from in-depth interviews with health workers, QCN district and national level coordinators as well as international stakeholders indicated non-reporting of activities on certain pillars meant such activities were not being carried out. It appeared that the theory of change was created at the outset of QCN but it is not that clear how much it guided actual activities or any monitoring and evaluation as things progressed. Enabling countries to develop their theory of change, perhaps guided by the global framework, could improve stakeholder engagement, progress to identify and work through assumptions relevant locally (e.g. regarding available infrastructure and data), and better target QCN work toward achieving its goals. Such work should be part of efforts to build learning health systems [39] that are able to create and sustain improved quality of care.

We propose that future efforts be directed toward the development of country-specific theories of change that capture the processes, causal pathways and assumptions in more detail and that those theories of change should be evaluated empirically.

## Supporting information

**S1 Text. PLOS GLOBAL HEALTH QCN evaluation collection 2-page summary.**
(DOCX)

**S2 Text. QCN papers common methods section.**
(DOCX)

## Acknowledgments

We thank all respondents and stakeholders for their time and contributions toward making this work possible. The QCN Evaluation Group is: Nehla Djellouli, Kasonde Mwaba, Callie Daniels-Howell, Tim Colbourn (UCL Institute for Global Health, UK), Kohenour Akter, Fatama Khatun, Mithun Sarker, Abdul Kuddus, Kishwar Azad (BADAS-PCP Bangladesh), Kondwani Mwandira, Albert Dube, Gladson Monjeza, Rachel Magaleta, Zabvuta Moffolo, Charles Makwenda (Parent and Child Health Initiative, Malawi), Mary Kinney, Fidele Mukinda (independent researchers, South Africa), Mike English (Oxford University), Yusra Shawar, Will Payne, Jeremy Shiffman (Johns Hopkins University, USA), Kathy Lubowa, Agnes Kyamulabi, Hilda Namakula, Gloria Seruwagi (Makerere University, Uganda), Anene Tesfa, Asebe Amenu, Theodros Getachew, Geremew Gonfa (Ethiopia Public Health Institute, Ethiopia), Seble Abreham, Tanya Marchant (LSHTM), UK

## Author Contributions

**Conceptualization:** Albert Dube, Kohenour Akter, Seblewengel Lemma, Gloria Seruwagi, Yusra Ribhi Shawar, Nehla Djellouli, Charles Mwakwenda, Mike English, Tim Colbourn.

**Formal analysis:** Albert Dube, Fatama khatun, Gloria Seruwagi, Nehla Djellouli, Charles Mwakwenda, Mike English, Tim Colbourn.

**Funding acquisition:** Yusra Ribhi Shawar, Mike English, Tim Colbourn.

**Investigation:** Albert Dube, Kondwani Mwandira, Kohenour Akter, Fatama khatun, Seblewengel Lemma, Nehla Djellouli, Tim Colbourn.

**Methodology:** Albert Dube, Kondwani Mwandira, Kohenour Akter, Fatama khatun, Seblewengel Lemma, Yusra Ribhi Shawar, Nehla Djellouli, Mike English.

**Resources:** Tim Colbourn.

**Supervision:** Kohenour Akter, Yusra Ribhi Shawar, Nehla Djellouli, Charles Mwakwenda, Mike English, Tim Colbourn.

**Validation:** Albert Dube, Kondwani Mwandira, Kohenour Akter, Fatama khatun, Seblewengel Lemma, Gloria Seruwagi, Yusra Ribhi Shawar, Nehla Djellouli, Mike English.

**Visualization:** Albert Dube.

**Writing – original draft:** Albert Dube, Tim Colbourn.

**Writing – review & editing:** Albert Dube, Kondwani Mwandira, Kohenour Akter, Fatama khatun, Seblewengel Lemma, Gloria Seruwagi, Yusra Ribhi Shawar, Nehla Djellouli, Charles Mwakwenda, Mike English, Tim Colbourn.

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
