## [Decision Letter · Decision Letter 0]

12 Feb 2024

PGPH-D-23-02391

Evaluating theory of change to improve the functioning of The Network for Improving Quality of Care for Maternal, Newborn and Child Health

Dear Dr. Colbourn,

Thank you for submitting your manuscript to PLOS Global Public Health. After careful consideration, we feel that it has merit but does not fully meet PLOS Global Public Health’s publication criteria as it currently stands. Therefore, we invite you to submit a revised version of the manuscript that addresses the points raised during the review process.

Please note that we have only been able to secure a single reviewer to assess your manuscript. We are issuing a decision on your manuscript at this point to prevent further delays in the evaluation of your manuscript. Please be aware that the editor who handles your revised manuscript might find it necessary to invite additional reviewers to assess this work once the revised manuscript is submitted. However, we will aim to proceed on the basis of this single review if possible. 

We look forward to receiving your revised manuscript.

Kind regards,

Johanna Pruller, Ph.D.

Staff Editor

Journal Requirements:

Additional Editor Comments (if provided):

Reviewers' comments:

Reviewer's Responses to Questions

**Comments to the Author**

1. Does this manuscript meet PLOS Global Public Health’s publication criteria? Is the manuscript technically sound, and do the data support the conclusions? The manuscript must describe methodologically and ethically rigorous research with conclusions that are appropriately drawn based on the data presented.

Reviewer #1: Yes

2. Has the statistical analysis been performed appropriately and rigorously?

Reviewer #1: N/A

3. Have the authors made all data underlying the findings in their manuscript fully available (please refer to the Data Availability Statement at the start of the manuscript PDF file)?

Reviewer #1: No

4. Is the manuscript presented in an intelligible fashion and written in standard English?

Reviewer #1: Yes

5. Review Comments to the Author

Reviewer #1: Thank you to the authors for the opportunity to review this insightful and well-presented paper. In the context of a large-scale evaluation in which multiple papers and studies contribute to rich understanding of a programme and its effects, it is often challenging to find the right balance in providing contextual background for individual studies. Some of my comments relate to this, and I hope are relatively easy to address.

I’ve divided my feedback into more major and minor comments/questions.

Major feedback:

I understand that many details about the QCN are undoubtably reported elsewhere, but reading the paper is occasionally challenging because of lingering questions about what exactly the QCN was and what was found in the other studies of the evaluation. One or two sentences about what the QCN did (for example, what were the “learning facilities”?) and what the overall evaluation of the network found would be very helpful to understanding this study’s results. This contextual background might fit well in the paragraph beginning line 90. Further, is there any background information about how the international and national partners developed their theory of change and how it was intended to be used (paragraph beginning line 78)?

Lines 69-75: The authors note that theories of change can have multiple applications, and the phrase is sometimes used with multiple meanings. Do the authors, or the creators of the QCN theory of change, take a particular stance on what the ‘theory of change’ is in this case? Figure 1 seems scant on some details, particularly around the assumptions that seem integral to a ‘proper’ theory of change and that are of core interest in this analysis. How did the authors go about investigating assumptions that may not have been stated explicitly in this diagram?

One suggestion for the paper is to better connect the three components of investigation – results, assumptions, and causal pathways (lines 124-127) – to the findings presented. And given the large number of assumptions and causal pathways that underlie such a complex programme, how did the authors approach the need to prioritise these within the analysis? The limitations note that not all areas of the theory of change could be investigated (lines 366-367), but how might the choices made in the analysis, or gaps in the data, leave limitations in our understanding of the programme?

Minor feedback:

Is it possible to add the job title/category to the respondent descriptors in the attributions of the quotes? If there are concerns about confidentiality of participants, even very broad categories (‘facility staff’ or ‘sub-national partner’) would help to contextualise the quotes presented.

Did none of the nine countries in the QCN develop country-level theories of change (or perhaps something else? impact pathways or logic models)?

Line 51: There’s a misprint here of the number of maternal deaths in 2020, by a magnitude of millions!

Lines 60-63: I strongly agree with the authors that interventions to improve quality of care “are complex and require comprehensive and detail [sic] theories of change”. However, I wonder whether it is the theory of change itself that contributes nuanced understanding for addressing evidence-to-practice gaps or is it the evaluations that are guided by these theories of change? (Good) evaluations interrogate assumptions and causal linkages embodied in theories of change, and we hope that this results in better design and implementation of programmes, which might be expressed in a more informative programme theory of change?

Line 99: Could the authors please provide more detail on who the “key stakeholders” included? Would these have included facility managers and/or staff on the frontlines of care? Ministry of Health officials? Others?

Line 115: Could the authors please add a citation to the adapted tool? This would be useful for others interested in conducting similar surveys.

Lines 175-178: This quote is quite difficult to follow as it is not clear who “they” refers to here.

Line 188: Should this be sub-national rather than national level initiative?

Lines 351-354: The first sentence here does not seem to follow with the rest of the paragraph, and it’s not clear how these points connect.

Line 362: Is it that community empowerment was not applicable or simply less of a priority?

Line 373: To what extent does not reported on = absent from activities?

6. PLOS authors have the option to publish the peer review history of their article (what does this mean?). If published, this will include your full peer review and any attached files.

**Do you want your identity to be public for this peer review?** For information about this choice, including consent withdrawal, please see our Privacy Policy.

Reviewer #1: **Yes: **Emma Radovich

---

## [Decision Letter · Decision Letter 1]

2 Jun 2024

PGPH-D-23-02391R1

Evaluating theory of change to improve the functioning of the network for improving quality of care for maternal, newborn and child health

Dear Dr. Colbourn,

Thank you for submitting your manuscript to PLOS Global Public Health. After careful consideration, we feel that it has merit but does not fully meet PLOS Global Public Health’s publication criteria as it currently stands. Therefore, we invite you to submit a revised version of the manuscript that addresses the points raised during the review process.

We look forward to receiving your revised manuscript.

Kind regards,

Jianhong Zhou

Staff Editor

Journal Requirements:

2. We noticed that you used "unpublished" in the manuscript. We do not allow these references, as the PLOS data access policy requires that all data be either published with the manuscript or made available in a publicly accessible database. Please amend the supplementary material to include the referenced data or remove the references.

Additional Editor Comments (if provided):

Reviewers' comments:

Reviewer's Responses to Questions

**Comments to the Author**

1. If the authors have adequately addressed your comments raised in a previous round of review and you feel that this manuscript is now acceptable for publication, you may indicate that here to bypass the “Comments to the Author” section, enter your conflict of interest statement in the “Confidential to Editor” section, and submit your "Accept" recommendation.

Reviewer #2: (No Response)

2. Does this manuscript meet PLOS Global Public Health’s publication criteria? Is the manuscript technically sound, and do the data support the conclusions? The manuscript must describe methodologically and ethically rigorous research with conclusions that are appropriately drawn based on the data presented.

Reviewer #2: Partly

3. Has the statistical analysis been performed appropriately and rigorously?

Reviewer #2: N/A

4. Have the authors made all data underlying the findings in their manuscript fully available (please refer to the Data Availability Statement at the start of the manuscript PDF file)?

Reviewer #2: Yes

5. Is the manuscript presented in an intelligible fashion and written in standard English?

Reviewer #2: Yes

6. Review Comments to the Author

Reviewer #2: Firstly, I would like to congratulate the authors for the beautiful research carried out. How important and indispensable it is for us to know the reality of care for the maternal and child segment, who are so fragile in relation to health care, especially in regions of such vulnerability. I also congratulate the authors for the brilliant initiative in evaluating whether the Network and the WHO proposal actually reached the places and people who really need it.

I am very pleased to participate in this stage and, in advance, know the results in countries that require attention across the globe.

My greatest concern is that, due to the magnitude of the work, the difficulties in presenting so much important data were perceived, and I fear that this may be perceived by readers as well. All the complementary material was extremely relevant for understanding the research, but I also understand when the authors mention the need for these materials to avoid overlapping information, since the entire set of results will be shown from 9 scientific articles. But this is a concern.

The introduction is concise and sufficient, and shows that there are problems within the scope of maternal and child health that require resolution, and that the proposed Network is an important step, but that the theory of change needs to be evaluated as well, to understand what worked and what didn't work, what are the challenges, needs, strategies for each scenario, propose and execute these changes and achieve quality and efficiency. So I have no suggestions.

Objective, clear and meets the research proposal.

Regarding the method, it is partially described, with the need to consult all the supporting material to understand how the research was conducted. This will need to be made very clear to the reader. And I ask, will all this material be available for consultation? What caught my attention was how the data collection was organized, who the researchers were, the number of researchers, how they made the first contact, what were the difficulties in accessing the participants and the documentary information, etc. This information was not described, not even in the supplementary material.

There were exclusion criteria for participants. Why were a greater number of interviews conducted in Bangladesh compared to other countries? At national and regional level? Since at the local level the number of interviews was equitable. Was there team training to conduct the interviews? What about recording all this material? How were the interviews and documents carried out?

The results were clearly described, with representations in participants' statements and a representative scheme of the main findings in each country.

The limitations of the study were exposed in a timely manner in the discussion, since the impossibility of showing concrete data, in terms of access, interventions, reduction or not of maternal and child morbidity and mortality could provide more accurate data on the practical application of the network and the proposed changes . And mainly whether or not the theory of change guided health actions in the countries involved. I congratulate the authors for raising this issue at the end of the article.

I only suggest that the authors reorganize the methodology, in order to provide more clarity to the reader on the way the study was conducted, without needing to consult all the complementary material.

7. PLOS authors have the option to publish the peer review history of their article (what does this mean?). If published, this will include your full peer review and any attached files.

**Do you want your identity to be public for this peer review?** For information about this choice, including consent withdrawal, please see our Privacy Policy.

Reviewer #2: No

---

## [Decision Letter · Decision Letter 2]

5 Jul 2024

Evaluating theory of change to improve the functioning of the network for improving quality of care for maternal, newborn and child health

PGPH-D-23-02391R2

Dear Prof Colbourn,

We are pleased to inform you that your manuscript 'Evaluating theory of change to improve the functioning of the network for improving quality of care for maternal, newborn and child health' has been provisionally accepted for publication in PLOS Global Public Health.

Best regards,

Julia Robinson

Executive Editor

Reviewer Comments (if any, and for reference):

Reviewer's Responses to Questions

**Comments to the Author**

1. If the authors have adequately addressed your comments raised in a previous round of review and you feel that this manuscript is now acceptable for publication, you may indicate that here to bypass the “Comments to the Author” section, enter your conflict of interest statement in the “Confidential to Editor” section, and submit your "Accept" recommendation.

Reviewer #1: All comments have been addressed

Reviewer #2: All comments have been addressed

2. Does this manuscript meet PLOS Global Public Health’s publication criteria? Is the manuscript technically sound, and do the data support the conclusions? The manuscript must describe methodologically and ethically rigorous research with conclusions that are appropriately drawn based on the data presented.

Reviewer #1: Yes

Reviewer #2: Yes

3. Has the statistical analysis been performed appropriately and rigorously?

Reviewer #1: N/A

Reviewer #2: N/A

4. Have the authors made all data underlying the findings in their manuscript fully available (please refer to the Data Availability Statement at the start of the manuscript PDF file)?

Reviewer #1: No

Reviewer #2: Yes

5. Is the manuscript presented in an intelligible fashion and written in standard English?

Reviewer #1: Yes

Reviewer #2: Yes

6. Review Comments to the Author

Reviewer #1: (No Response)

Reviewer #2: (No Response)

7. PLOS authors have the option to publish the peer review history of their article (what does this mean?). If published, this will include your full peer review and any attached files.

**Do you want your identity to be public for this peer review?** For information about this choice, including consent withdrawal, please see our Privacy Policy.

Reviewer #1: No

Reviewer #2: No
